# An Appraisal of Antidotes’ Effectiveness: Evidence of the Use of Phyto-Antidotes and Biotechnological Advancements

**DOI:** 10.3390/molecules25071516

**Published:** 2020-03-26

**Authors:** Christiana Eleojo Aruwa, Yusuf Ola Mukaila, Abdulwakeel Ayokun-nun Ajao, Saheed Sabiu

**Affiliations:** 1Department of Biotechnology and Food Technology, Durban University of Technology, P.O. Box 1334, Durban 4000, South Africa; 2Department of Botany, Obafemi Awolowo University, Ile-Ife 220005, Nigeria; 3Department of Botany and Plant Biotechnology, University of Johannesburg, P.O. Box 524, Auckland Park APK 2006, South Africa

**Keywords:** antidotes, phyto-antidotes, poisons, secondary metabolites, snakebites, snake envenomation

## Abstract

Poisoning is the greatest source of avoidable death in the world and can result from industrial exhausts, incessant bush burning, drug overdose, accidental toxication or snake envenomation. Since the advent of Albert Calmette’s cobra venom antidote, efforts have been geared towards antidotes development for various poisons to date. While there are resources and facilities to tackle poisoning in urban areas, rural areas and developing countries are challenged with poisoning management due to either the absence of or inadequate facilities and this has paved the way for phyto-antidotes, some of which have been scientifically validated. This review presents the scope of antidotes’ effectiveness in different experimental models and biotechnological advancements in antidote research for future applications. While pockets of evidence of the effectiveness of antidotes exist *in vitro* and *in vivo* with ample biotechnological developments, the utilization of analytic assays on existing and newly developed antidotes that have surpassed the proof of concept stage, as well as the inclusion of antidote’s short and long-term risk assessment report, will help in providing the required scientific evidence(s) prior to regulatory authorities’ approval.

## 1. Introduction

Poisons are substances that cause injury to or death of living tissues when swallowed, injected or inhaled. Unintentional poisoning or self-poisoning in addition to envenomation by several species of snakes are major causes of death in the world, causing thousands of deaths every year [1]. Globally, 20 persons are estimated to die from toxins/poisons every 40 minutes [2,3]. In the United States of America alone, one poisoning case is reported every 14.9 seconds, of which 9% are said to be critical [4]. Similarly, over 3000 poisoning deaths are reported yearly in Korea [5]. Further, the recent amazon fires sent thousands of citizens of the surrounding countries to hospital due to carbon monoxide poisoning and it was reported that German hospitals treated 205,000 acute poisoning patients in 2011 [6]. Annually, more than 125,000 deaths have also been attributed to snake envenomation [7]. These estimated figures are likely higher as only less than 5% of poisoning deaths are reported to poison control centres [8]. While snake bite poisoning occurs more in underdeveloped and rural areas, poisoning mortality in the developed world is caused majorly by chemicals and drug overdose as well as accidental poisoning in children [9].

Antidotes/neutralizers/antitoxins/nostra/antisera are substances with properties that can be harnessed to reverse, counteract, treat or manage a poisoned state [10]. They could also be any therapeutic substance used to abate the toxic effects of a specified xenobiotic over a period of time [11]. They are the general remedy to poisons and their application is as old as medicine itself. The role of antidotes is therefore vital in emergency and supportive care or as first aid prior to the initiation of full course treatments. In other words, and depending on the prevailing circumstances, antidotes may help mitigate dire consequences from poisoning, reduce mortality and morbidity as well as medical and non-medical requirements for patients care [10]. Certain antidotes are usually said to have universal functions—that is, “universal remedies or panacea” for poisons/toxins of a chemical or biological nature. A good and utilizable antidote is expected to be cheap, safe and readily and widely available. It must also have proven effect(s), an ideal dose, as well as a rapid and simple diagnostic test to determine the causative poison/toxin and extent of poisoning [2]. A good antidote may also only require little or no expertise for administration, not need high level healthcare infrastructure and, importantly, must have little or no adverse effects when administered under specific conditions. These properties are particularly important in remote, underdeveloped areas where access to healthcare facilities is limited or non-existent [2]. Antidote use and stocking are, however, subject to certain national and international guidelines.

The British Association of Emergency Medicine (BAEM) 2006 rules outline suitable ways and areas to keep antidotes in healthcare facilities, as well as their sufficient quantities. In line with this, antidotes are grouped into immediate, within an hour, within four hours and not critically time dependent classes, based on clinical urgency for use [12]. It is, however, vital to mention that in some situations, antidotes are hard to obtain, especially in remote areas where they are not available as pharmaceuticals. Also, the contraindications and efficacy of many antidotes remains contentious. Antisera efficiency also varies widely [13,14]. The importance of appropriate and timely use of antidotes cannot be overemphasized because these also impact their efficacy. Improper antidote usage and storage may result in little or no benefit to the patient or even harm from the antidote itself or the poison it is meant to counter [15,16]. Some antitoxins are slow, others may be fast acting, while others may only mitigate some of the toxic impacts of specific compounds. The relative amount of the antisera present compared to that of the toxic/poisoning agent also largely determines the efficacy of an antidote [13]. Nonetheless, while the evaluation of these significant parameters remains highly essential, the global availability and use of known and acceptable antidotal substances needs to be encouraged. Availability varies from nation to nation depending on several related categories, for example, time/geospatial, technical/scientific and economic and administrative and regulatory demands [13]. Furthermore, while antidotes are hoarded for disaster incidents such as chemical terrorist occurrences, rises in prices and new shortfalls have been created which worsen and deepen the challenges linked with providing access to antidotes for patients. Hence, rather than reduce deaths, such a situation may invariably result in increased mortality from poisoning [17,18]. Antidotes were reportedly used in United States (US) poison centres in 184,742 incidents in 2015 alone [15,19]. However, relevant antisera not being stocked and in insufficient amounts remains a daunting problem. This problem has been repeatedly documented for a range of antidotes in many developed nations, including the United Kingdom (UK), Canada, the US [20,21], Qatar [22] and Korea [5].

While antidotes’ effectiveness is studied, information on their use, storage and administration is too often not addressed [23]. For instance, the ideal and specific time for antidote delivery is crucial, yet only a handful of investigations adequately incorporate this in their study design. Again, when mention is made on an antidote’s early use, they rarely add exhaustive research data to support the claim. Few reports are also empowered to investigate the effect of delay in antidote management [15,24]. In addition, although nostrum development has been opined to proceed in the usual scientific flow from observation, experimentation, safety assessments and then clinical trials, this sequence is sometimes diverged from. This was the case involving the development of fomepizole to remedy toxic alcohol overdose, as the first article on its efficacy and safety was only published 18 years later in 1999 after its first clinical use. Such reversal in the process of translational research which begins from beside and ends on the bench should be strongly discouraged [25,26]. Therefore, the constant review and update of research information available on old and new antidotes remains essential. To date, several chemical compounds have been developed as antidotes against poisons of different origins while there are many botanicals used indigenously against poisoning, particularly snake venom poisoning. Even though there have been previous reviews on the use of herbal remedies for snake poisons [27,28] and other poisons [29], there have been no comprehensive reviews on the status of antidotes’ effectiveness in treating all cases of poisoning as well as recent developments in this field. In view of the foregoing, this article appraises the effectiveness of existing antidotes in *in vivo* and *in vitro* investigations. It also highlights antidote sources, classes, modes of action, the relevance of translational research by identifying valuable plants species used as antidotes, as well as relevant issues surrounding biotechnological developments in antidote research for future applications.

## 2. Materials and Methods

A literature search was conducted using various electronic data pools such as PubMed, Google Scholar, Scopus, MeSH, ScienceDirect and other reputable scientific sites. Search words and phrases that were used were relevant to the scope of the review and surrounded the subject of antidotes, their availability and limitations attributed to their global use, antidote classes and their mechanisms of action, snake venom, antisera/antitoxins of plant origin, *in vivo* and *in vitro* research on antidote utilization, from their respective inceptions up to December 2019 in an effort to streamline sought outcomes for the appraisal of antidotes’ efficacies in experimental studies. Specific highlights associated with biotechnological developments related to antidotes use and delivery/administration were also included. All information gathered on plants and substances with antidotal properties were utilized to generate a mechanistic model as depicted in Figure 1. The figure outline includes a basis for inclusion or exclusion of research records which are relevant to the topic under review.

## 3. Results and Discussion

### 3.1. Classes of Antidotes and Mechanism(s) of Action

Specific antidotes are developed to counteract the adverse effects of specific poisons, therefore there are as many antidotes as poisons. There are no general classifications of antidotes as different authors have used different classifications. For instance, antidotes have been classified based on their documented efficacy [5], mechanisms of action [29,30], the group of poisons they are used against [31] and based on clinical urgency of use [12].

Based on their mechanisms of action, antidotes are classified as discussed below.

#### 3.1.1. Competitive Antagonists

Antagonists are chemical substances (drugs) that binds to receptors without producing a notable stimulation of the receptor. This is the most common mechanism where the antidotes bind reversibly to cellular receptors, competing with and ultimately displacing the poisons from binding with active receptors, thereby reducing the amount of effective poisons. For example, while vitamin K, an antidote for anticoagulant poisoning, competes with the poison at the active site of production of prothrombin in the liver, naloxone, an antidote for most narcotic analgesics poisoning like heroin, competes against the poison molecules at the opioid receptor site [29,30].

#### 3.1.2. Chelating Agents

These are antidotes that react with the poison to form an inert complex which is not immediately harmful to the body and is later removed from the body through excretion. Examples include most metal (platinum, iron, cadmium, copper, mercury, aluminium, lead, nickel, arsenic, etc.) poisoning antidotes, for example, 2, 3-dimercaptosuccinic acid (DMSA) for lead poisoning, dicobalt edetate for cyanide poisoning and Prussian blue for thallium poisoning [29,30]. Other known chelators include dimercaprol (BAL), N-acetyl Cysteine (NAC), unithiol (DMPS), D-penicillamine (DPA), zinc trisodium or calcium trisodium diethylenetriaminepentaacetate (ZnNa_3_DTPA/CaNa_3_DTPA), N-acetyl-D-penicillamine (NAPA), deferoxamine (DFO), calcium disodium ethylenediaminetetraacetate (CaNa_2_EDTA), triethylenetetraamine (trientine) and deferiprone (L1). Synthetic analogues which have been tested *in vivo* include carbodithioates, BAL derivatives (mono- and dialkylesters of DMSA) and polyaminopolycarboxylic acids (EDTA and DTPA) [32,33]. Generally, all metal chelators possess free electrons which bind positively charged ions transition metal ions by forming a complex with two or more chelate rings. The complex is then transformed with biological ligands into a new and less harmful complex that is passed out from the organism. Good chelators must be easily absorbed from the gastrointestinal tract (GIT), show minimal toxicity, low affinity for essential metals within the organism it is administered to and high affinity for the poisoning metal [33]. Just as the poisonous impact of each heavy metal varies, the pharmacokinetics, side effects and clinical application of each chelator also differs widely [33].

#### 3.1.3. Acceleration of Detoxification

Some antidotes help to speed up the process of conversion of the poison to a non-toxic substance—a typical example is the action of acetylcysteine in accelerating the detoxification of potentially toxic metabolites during paracetamol poisoning while thiosulphate accelerates the conversion of cyanide to thiocyanate, a non-toxic substance [29,30]. Paracetamol poisoning from overdose and abuse is common and closely linked with drug-induced liver failure [34]. Compared to methionine and cysteamine, parvolex or intravenous NAC has been shown to be the most effective antidote for paracetamol poisoning which when dispensed appropriately, can prevent paracetamol-related nephro- and hepato-toxicity [35]. NAC is a glutathione donor that acts in the liver by the conjugation of glutathione with N-acetyl-P-benzoquinoneimine [12]. In terms of side effects, certain patients show anaphylactoid response characterised by upregulated levels of histamine, believed to be released from basophils in circulation [36,37].

#### 3.1.4. Reduced Toxicity

There are antidotes that reduce the toxicity of poisons by either facilitating the conversion of the poison to a less toxic substance or inhibiting the conversion to a potentially toxic substance. For example, in methaemoglobin poisoning, methylthionium chloride is a cofactor for the reduction of the methemoglobin by NADPH, while in methanol poisoning, ethanol inhibits the conversion of methanol to toxic metabolites by competing with the methanol for alcohol hydrogenase, the enzyme that is required for the process [29,30].

#### 3.1.5. Receptor Site Blocker

Some antidotes bind to a receptor site before the toxin, thereby blocking the site and preventing the toxin from binding—for example, atropine used in organophosphate poisoning (OP) blocks the muscarinic receptor site and prevents the poison from binding and exerting its effect [29]. Similarly, while anexate functions as a γ-aminobutyric acid (GABA) receptor which can reverse benzodiazepine sedative effects induced from intensive care or anaesthesia [12], fomepizole is an atypical alcohol dehydrogenase blocker that could reverse toxic alcohol (methanol and ethylene glycol) poisoning [26].

#### 3.1.6. Cyanide Binders/Sulphur Donors

Cyanide antidotes are categorised into physiological, biochemical, as well as detoxification and scavenging mode of action [38]. Oxygen is a physiological cyanide antiserum which acts by improving sulphite oxidation, which invariably facilitates cyanide detoxification. On the other hand, biochemical cyanide antidotes are largely non-specific with unclear modes of action and may therefore act as adjuncts to other agents/antidotes. For example, chlorpromazine is believed to prevent cell membrane lipids peroxidation, sustain cellular calcium balance and upregulate the efficiency of combined sodium nitrite (SN) and sodium thiosulphate (STS) in cyanide toxicity. Other examples include papaverine, phenoxybenzamine (an α-adrenergic blocking agent), antihistamines, antioxidants and nitric oxide generators [38,39]. Cyanohydrin and methaemaglobin formers and cobalt-containing compounds are grouped under scavengers. Methaemaglobinaemia (MetHB) formers such as amyl, nitrite, SN; cobalt compounds like hydroxocobalamin (cyanokit), dicobalt edetate; and sulphur donors such as STS are useful antidotes to combat cyanide poisoning, but usage is dependent on blood cyanide concentration. MetHB formers bind cyanide to form an inactive cyan-methaemoglobin compound. Dicobalt edetate form less toxic cobalticyanide and cobalto-cyanide ions after combining with cobalt [12]. STS falls under the cyanide detoxifying class. It enzymatically converts mitochondrial rhodanase to thiocyanate. This conversion occurs slowly through the production of sulphane sulphur which binds cyanide to produce less harmful thiocyanate. STS is also useful as an adjunct to other agents in cases of severe toxicity or in moderate toxicity cases with minimal adverse effects [38,40].

#### 3.1.7. Cardiac Drug Antidotes

An overdose of calcium channel blocker such as diltiazem, verapamil and nifedipine lead to cardiovascular toxicity. This may be remedied by the use of calcium chloride or calcium gluconate. These salts act by upregulating cellular calcium levels. In the case of beta blockers (atenolol, propanolol) overdose, glucagon, occasionally followed by an anti-emetic, is the antidote of choice. Glucagon acts by stimulating adenylate cyclase independently of beta-receptors [12].

#### 3.1.8. Universal/General Antidotes

In some circumstances, the swift reversal of anticoagulation may be required orally. Where this is related to non-vitamin K or vitamin K antagonists, prothrombin complex concentrate (PCC), such as warfarin and phenprocoumon coumarins, have proved effective. These coumarins act by reinstating the active vitamin K form by inhibiting vitamin K reductase [41,42]. Dosage in form of an international normalized ratio (INR) which serves as a guide remains a subject of debate and needs close monitoring [43]. However, a new class of oral, non-vitamin K anticoagulants which do not require much monitoring or dose adjustment include dabigatran, rivaroxaban, edoxaban and apixaban. Again, novel molecules considered to be safest and most efficacious such as idarucizumab, PER977 and andexanet still remain in premarketing studies [44,45]. Intravenous lipid emulsions (ILEs) are another example of wide spectrum antidotes utilized in cases of cardiotoxicity [46]. Activated charcoal is also usually called a universal antidote as it is indicated for most poisons and acts by decreasing the absorption of potentially harmful poisons in the GIT. Activated charcoal prevents the absorption of poisons by the stomach and intestine by binding the poison and interrupting enterohepatic circulation, though it must be administered within the first hour of the ingestion of poison for this to be possible. [9]. Despite its efficacy, the use of activated charcoal has gradually declined to about 5.6% in 2004 from 7.7% in 1995 because it is contraindicated in certain situations and strongly cautioned in others. It is also not useful in poisonings related to heavy metals, corrosives, alcohol and hydrocarbons [47]. In summary, Table 1 gives examples of selected antidotes and their respective indications.

### 3.2. In vitro Studies on the Use of Antidotes

*In vitro* models provide a solid basis for the evaluation of the efficacy of an antidote because it bars inter/intra-species differences in *in vivo* experiments and could mimic actual biological scenarios [48]. An *in vitro* study using human whole blood and plasma surrogate samples was conducted to confirm the effect of DMSA on mercury poisoning and found that it substantially abated the effect of mercury poisoning even though their study suggested that the word “chelation therapy” is inappropriately used for DMSA remediation of mercury poisoning because there was no formation of chelation complexes as suggested by previous studies [49]. A deliberate poison of great concern is the various organophosphorus compounds (nerve agents) like soman, tabun and cyclosarin gases which when inhaled, inhibit human acetylcholinesterase (AChE), causing swift death and its antidotes are the various bispyridinium oximes that reactivate the inhibited AChE [50]. An *in vitro* study was conducted to evaluate the potency of bispyridinium oxime K203 [(E)-1-(4-carbamoylpyridinium)-4-(4-hydroxyimino methylpyridinium)-but-2-ene dibromide] and found that it is a very potent reactivator of tabun-inhibited human and protected against phosphorylation of AChE by tabun [51]. Ethylenediaminetetraacetic acid (EDTA), meso-2,3-dimercaptopropane-1-sulfonic acid (DMPS), diethylcarbodithiolate (DDTC), Vitamins B1 (thiamine), B6 (pyridoxine) and C (ascorbic acid) were found to inhibit lead uptake and reduced lead cytotoxicity in lead poisoning as opposed to vitamins B2 (riboflavin) and B12 (cobalamine), which were ineffective [52]. Similarly, the *in vitro* efficacy of oximes as antidotes for acute OP is known [53]. This is in contrast with reported oxime treatment failure in human trials which indicated that oximes may be ineffective or harmful [54]. Such studies may, however, have had apparent technical challenges which did not allow true measurement of oximes efficacy in patients presenting with acute OP symptoms [2,55]. Compared to methylpyridinium aldoxime antidote, ionized zwitterionic aldoximes, an alternative and highly functional OP antidote with both therapeutic and prophylactic effects, has shown greater promise for use in tackling OP nerve agents like OP-based pesticides and sarin [56,57].

Furthermore, the dose of the oxime chosen was often not related to the species-specific effect of the antidote for reactivation of inhibited AChE, but frequently to the toxicity of the oxime. Certainly, an oxime alone without an antimuscarinic shows only marginal efficacy [53] as it is not lipophilic and does not readily penetrate into the central nervous system (CNS). From the fact that effective treatment of OP must include countermeasures against muscarinic and nicotinic over-stimulation, it is evident that the dose of atropine is very important for the estimation of oxime efficacy [58]. In a more recent study, obidoxime, an antidote for paraoxon, an organophosphate toxin, was reported to impact blood coagulation and inhibiting thrombin function *in vitro*, thus indicating its potential application in blood coagulation pathways [59]. Lovrić et al. [60] also demonstrated the therapeutic impact of a newly synthesized compound, atropine-4-pyridinealdoxime (ATR-4-OX) linked to the reactivation of phosphorylated AChE, as against receptor antagonization. *In vitro* investigations were carried out on paraoxon-inhibited AChE of human erythrocytes [60]. In another study by Kalagatur et al. [61], *Jatropha curcas* seed shells from which activated carbon was derived (through zinc chloride activation technique) significantly detoxified zearalenone mycotoxin in *in vitro* neuro-2a cells. This indicated that activated carbon obtained from *Jatropha curcas* may be successfully utilized as an antidote for the decontamination of zearalenone-induced toxicity. The potential applications in food poisoning require further exploration [61]. Another study synthesized a cationic polysaccharide-based (dextran substituted with glycidyltrimethylammonium chloride groups—Dex40-GTMAC3, which was most potent) molecule/antidote with enhanced unfractionated heparin (UFH) uptake that is as efficient as protamine, but safer with lower immunogenicity and production cost. Until now, protamine had been the only antidote for UFH [62].

In certain scientific terms, it may be surmised that dose-impact relationship studies using systematic *in vitro* models like human materials and in-depth assessment of analogous human poisoning provide evidence of antidote therapeutic efficacies. It also makes up for situations where human clinical studies are lacking, especially when drug approval is required by regulatory agencies. An added advantage is the protection of animal life and prevention of invasive or painful procedures which animals/humans are subjected to [63].

### 3.3. In vivo Studies On The Use Of Antidotes

There have been several *in vivo* studies on the effectiveness of antidotes owing to the alarming increase in the number of deaths caused by poisoning, though there has always been a grey area of difference in response to antidotes and poisons among different species [48]. For example, great differences were found in the potencies of oximes in human and animals (guinea pigs, rats and rabbits) in an experiment to determine the reactivation rate constant of erythrocyte AChE inhibited by sarin, cyclosarin and VX [64]. NAC was found to induce excretion of between 91% and 94% of methylmercury poison initially administered to a set of mice, though there were no significant difference between the mercury content in the faeces of mice that received the NAC and the control. It was also noticed that NAC successfully removed mercury poisons from all examined tissues including brain tissues [65]. Even though sodium nitrite (SN) and sodium thiosulphate (STS) have been proven to be an effective antidote of cyanide poisoning [66], it was showed in an *in vivo* experiment using Wistar rats that α-Ketoglutarate (α-KG) could be a viable alternative as it was reported to be safer, detoxifies faster and can be administered orally as opposed to the intravenous administration of the SN and STS [67]. For the nerve agent tabun poisoning, a quarter of the LD_50_ of K203 combined with atropine protected test mice against all toxic effects associated with the poison and all test organisms survived, even at a poison dose of 8.0 LD_50_ [51]. Similarly, in an *in vivo* experiment with rats and soman gas, it was reported that a quarter of the LD_50_ of HI-6 [(1-(2-hydroxyiminomethylpyridinium)-3-(4-carbamolypyridinium)-2-oxapropane dichloride)] combined with atropine substantially protected all experimental animals against a soman dose of 3.2 LD_50_ [50]. In the chelation therapy for heavy metal poisoning, data from 600 patients with varieties of complaints but not acute metal poisoning were evaluated and it was found that both calcium and sodium EDTA are equally effective in the removal of lead, while calcium EDTA is more effective in the removal of aluminium. The removal of these metals was found to be dose-related but not in a linear relationship and it was concluded that to avoid the risk of antidote poisoning, low dosage of chelating agents are preferable [68].

Cisplatin (platinum-based compound) is a known molecule for the treatment of testicular cancer of which analogues have also recently been shown to function in cyanide detoxification. *In vivo* cisplatin analogues demonstrated protection against the harmful effects of cyanide in mice, zebra fish and rabbit models. Using optical spectroscopy, the analogues administered intravenously also reinstated cytochrome C oxidase redox state in rabbit muscles, making it an effective mammalian antidote [69]. Reports have demonstrated that higher mammals such as humans, monkeys and guinea pigs are better suited for toxicity studies on nerve agents [70]. This was attributed to significant carboxylesterase concentrations (like levels in humans) which scavenges OP compounds in laboratory models such as rats and mice [70]. These were, therefore, used in OP compounds’ antidote evaluation [71]. Oximes also show species-specific variation in their ability to reactivate OP inhibited AChE [63,64].

A squalene-based nanoantidote (squarticles with diameter range of 97 to 122 nm) has also shown promise for the reversal of amitriptyline (an antidepressant) intoxication [72]. Anionic squarticles were shown to be significantly efficacious in sequestering and trapping the drug and improving the survival rate in rat models induced with amitriptyline intoxication by up to 75%. It also had a more than two-fold increase in plasma drug concentration and long-term retention *in vivo*. The potent ability of this novel nanoantidote to eliminate amitriptyline, a tricyclic antidepressant (TCA), in animal models shows great promise, especially for intensive care units (ICU) and ICU patients. Further studies are still required on the effect of squarticles on other TCAs [72]. Another study on a novel, automated and rapid antidote (naloxone) delivery device (A2D2) showed reduction in response time for delivery of relatively significant amounts of the antisera in mice subcutaneous tissues. Still required, however, are long term stability studies, as well as the release mechanism required to revive overdosed animal models [73]. In addition, broad spectrum Intravenous Lipid Emulsions (ILEs) antidotes such as ClinOleic (Baxter) and Liposyn (Hospira) have been produced and used to rescue a cardiac arrest patient for the first time [74] after its potential as an antidote to reverse cardiotoxicity in rats was demonstrated eight years earlier [75]. The resuscitative function of ILEs has been optimally utilized ever since being used in intoxication cases related to cardiovascular collapse [76]. ILEs were, therefore, considered as the most investigated antidote treatments in toxicology [77]. For cyanide-related poisoning, the intramuscular injection of dimethyl trisulfide (DMTS), a new cyanide antidote, has been shown in a cell culture blood-brain barrier (BBB) and mice models to significantly be diffused with a 13 times increase in permeability through the cell culture [78]. Also, ethotoin, a dose-dependent ricin toxin inhibitor, has been shown to reduce the mortality rate in mice models injected via the intranasal and intra-peritoneal routes with lethal doses of ricin, thus showing potential for the reversal of food intoxications and/or bioterror circumstances linked to ricin [79]. More recently, cobalt-containing macrocyclic (CCM) antidotes with established efficacy against cyanide toxicity were also found to attenuate sublethal azide toxicity in male and female Swiss-Webster mice [80]. The lowest effective doses of the CCM antidotes remarkably shortened the recovery time of the azide-poisoned mice as elucidated in a pole climbing experimental model [80]. A list of selected studies lending credence to the use and effectiveness of common antidotes in both *in vivo* and *in vitro* experimental models is presented in Table 2.

### 3.4. Snake Venoms As Poisons

Generally, snake venoms are classified based on how the effect of their principal toxins are exerted, and three main classes have been identified: Cytotoxins (e.g. cardiotoxins, myotoxins, nephrotoxins), hemotoxins and neurotoxins [81,82,83].

#### 3.4.1. Cytotoxins

Cytotoxic venoms kill cells and tissue and, in some cases, liquefies the tissue, partially digesting the organism before it is eaten. The ability of snake venom cytotoxins to kill cells and tissues have been said to be a valuable property in the treatment of tumour cells [84] as well as cancer [83], for example, ACTX-8 isolated from the venom of *Agkistrodon acutus* caused apoptosis in Hela cervical cancer cells [85]. Cytotoxins are basically toxic β-structured proteins consisting of 59-61 amino acid residues and they constitute about 40%-70% of the venom of cobra snakes in the genus *Naja* and *Haemachatus* [86]. *Haemachatus* are found mainly in Africa, distributed across South Africa, Swaziland and some parts of Mozambique and Zimbabwe. *Naja* is a large genus of cobras distributed throughout the tropical and subtropical regions of the world from Africa to Asia. Cytotoxins are sometimes specific in the type of cells or tissue they attack. For example, cardiotoxins are cytotoxins that attack the heart tissues, myotoxins attack the muscle tissues, while nephrotoxins inflict the cells of the kidney [82].

#### 3.4.2. Hemotoxins

These are groups of venoms that poison the blood, disrupting normal blood clotting, blood pressure, coagulation and circulation processes. Components of hemotoxins are snake venom metalloproteinases (SVMPs) and PLA_2_s [81]. These toxic substances rupture red blood cells and cause internal bleeding. In some other cases, some hemotoxins cause components of the blood to clot, leading to blockage of blood vessels and disruption in circulation. In general, hemotoxins are either procoagulant, causing the blood to clot within the vein, or anticoagulant, affecting the ability of blood to clot [87]. Despite this, many hemotoxins have been said to have potential therapeutic effects and some can be used as laboratory reagent [88]. Serpents of the family Viperidae are well known to produce many hemotoxins [82]. Vipers are native to the United States of America, Mexico, Central and South America but are found throughout the world except for Antarctica, Australia, New Zealand, Hawaii and Madagascar.

#### 3.4.3. Neurotoxins

These group of toxins affect the nervous system by strongly binding with nicotinic acetylcholine receptors (nAchRs) [82]. Most neurotoxins are composed of the three-finger toxins (3FTx) and phospholipases A_2_ (PLA_2_s) [89]. They inhibit communication between neurons across the synapse, causing muscle paralysis and resulting in respiratory difficulties and subsequent swift death. Snakes in the Elapidae family typically produce these group of toxins. Typical examples of neurotoxins are highlighted and discussed below.

##### Calciseptine

This neurotoxin is produced by *Dendroaspis polylepis* (Black mamba). It disrupts the transduction of nervous impulse by blocking voltage-gated calcium channels [82]. Calciseptines have been shown to have several uses in medicine, for example, it has hypotensive activity [90] and can induce muscle relaxation [91]. Black mambas are found in most sub Saharan African countries from South Africa to Cameroon, Ethiopia, Eritrea and Congo.

##### Cobrotoxins

These toxins block nicotinic acetylcholine receptors (nAchRs), resulting in paralysis [82]. It is composed of 15 amino acids in a single peptide chain linked intermolecularly by four disulphide bonds [92]. They are produced by snakes of the genus *Naja* (Cobras).

##### Calcicludine

They disrupt nerve signals by blocking voltage-gated calcium channels (L-, N and P-type) [82,93] and are produced by *Dendroaspis angusticeps* (Eastern green mamba), which are native to eastern Africa from south Kenya to Tanzania, Malawi, eastern Zimbabwe and Zambia.

##### Fasciculin-II

Also produced by *Dendroaspis angusticeps*, inhibits acetylcholinesterase (AChE), resulting in convulsion and paralysis [82].

##### Caliotoxins

They are produced by *Calliophis bivirgatus* (Blue coral snakes). They attack sodium channels, resulting in paralysis [82]. *Calliophis bivirgatus* is native to South East Asia and are found in Brunei, Indonesia, Malaysia, Singapore, Thailand and Burma.

##### Bungarotoxins

This toxin is produced by *Bungarus caeruleus* and paralyses the muscles by attacking nerve endings near the synaptic cleft of the left brain, resulting in severe abdominal pain, breathing difficulties and death. *Bungarus caeruleus* is found in south India, Sri Lanka, Pakistan, Bangladesh and Nepal [82,94].

### 3.5. Natural antidotes of plant origin

#### 3.5.1. Evidence of use and effectiveness against snakebite and other animals’ poisons

Poisonous insects, spiders, annelids, scorpions and venomous snakes are examples of common toxin-producing animals. In Assam, India, irritations from caterpillar stings are relieved using a paste from *Kaempferia galangal*, while *Hibiscus sabdariffa* or *Lagenaria siceraria* (Molina) are used as antidotes for ant bites and bee stings. Rhizome paste of *Boesenbergia rotunda* (Linnaeus) with analgesic and anti-inflammatory activities serves as an antidote for catfish stings, while *Allium sativum* paste is applied on spider bites [95].

Specifically, snakebite, recently categorized as a Neglected Tropical Disease (NTD), is a serious health issue in tropical and subtropical regions of the world [96]. There have been several reports on the use of herbal antidotes against snake poisons and some initial scientific reports concluded that the majority of these remedies were ineffective [97]. However, subsequent studies have challenged those findings and suggested consideration of systemic changes accompanied by the venoms in their experimental models and validating the folkloric uses of many antidotal plants in the process [27]. The following studies have provided scientific credence to the use of plants in remedying snakebites; 12 plants initially reported to be used against snakebites in Columbia [98] were screened for their potency and anti-snake venom activities against “fer-de-lance” (*Bothros asper*) venom. Ethanolic extracts of all the plants showed substantial activities with six of them (*Bixa Orellana*, *Brownea rosadenonte*, *Dracotium croati*, *Gonzalagunia panamensis*, *Struthanthus orbicularis* and *Trichomanes elegans*) completely inhibiting the negative effect of the venom [99]. In an *in vivo* experiment using rats, extracts from the rhizome of *Curcuma longa* inactivated the neurotoxins of *Naja naja siamensis* [100], while aqueous extract of *Withania sonnifera* also completely neutralized the PLA_2_ of *Naja naja* venom [101]. Others include *Vitex negundo* and *Emblica officinalis* [102], *Hemidesmus indicus* [103,104], *Crinum jagus* [105], *Strychnus nux vomica* [106], root paste of *Mirabilis jalapa*, *Pogostemon purviflorus* and *Rauwolfia serpentina*, *Acorus calamus* and *Kaempferia galanga* rhizome, *Dioscorea* sp. tuber paste and leaf paste of *Xanthium strumarium* leaves [95]. Although studies have focused on the potency of herbal antidotes of folk medicine and their bioactive constituents, only few have paid attention to the mechanism of the antidotal effect of these plants. Nonetheless, few hypotheses have been put forward. For example, it was concluded from both *in vitro* and *in vivo* experiments that melanin extracted from black tea (*Thea sinensis* L.) was able to inhibit the PLA_2_ activity of snake venoms because it was able to chelate Ca^2+^ which is an important co-factor of PLA_2_ even though the effect was not quantified. The study also proposed that this inhibitory effect could be due to direct binding of PLA_2_ by the melanin extract but a comparison with p-bromophenacyl bromide, a specific binder of PLA_2_, showed a significant difference, prompting the conclusion that the melanin extract is a non-specific binder of PLA_2_ [107]. Aqueous extract from the leaves of *Schizolobium parahyba* completely neutralized the lethal effect of *Bothrops alternatus* and *Bothrops moojeni* venom, and the mechanism was thought to be precipitation of the active protein in the venom but subsequent fractionation showed that the fraction that inhibited 100% of the venom did not exhibit protein precipitation while the fraction that showed protein precipitation did not inhibit the venom [108]. It was proposed that the antivenin activity of 2-hydroxy-4-methoxy benzoic acid extracted from the root of *Hemidesmus indicus* against the venoms of *Vipera russellii, Naja kaouthia, Ophiophagus hannah* and *Echis carinatus* was due to adjuvant effects by triggering the production of antibodies capable of neutralizing the effects of the venoms [109], corroborating a previous study where the same compound activated lymphocytes and macrophages in male albino mice [110]. Lupeol acetate also extracted from *Hemidesmus indicus* was observed to neutralize *Daboia russellii* and *Naja kaouthia* venom and the mechanism was thought to be its strong antioxidant property which mitigated the inflammatory activity of the venom, though no clear mechanism was reported [111].

Generally, investigations into ethnomedicinal plants have demonstrated the presence of saponins, tannin, alkaloids, sulphur, anthraquinones, terpenoids, oleanolic acid, flavonoids, mannitol, glycosides, urosolic acids, pyrrolidine, formic acids, essential oils and steroids bioactive components [112,113]. These compounds are plant secondary metabolites that may act singly or synergistically to bring about their antidotal effects. In fact, it has been opined that the potent ability of plants to detoxify poisons is largely hinged on the presence of these metabolites [95]. Medicinal plant extracts produce a combined effect since they contain a mix of active components of which their total activity surpasses that of purified and separated individual components as is the case with most pharmacological drugs. The presence of these plant metabolites reduces adverse side effects, enhances active phytochemicals stability and have potentiating, antagonistic or additive effects as well [114]. The pharmacological study of plant-based folk antidotes is a worthwhile venture for ascertaining the active ingredients, as well as the mechanism of action of folkloric plants with medicinal properties [95].

#### 3.5.2. Evidence of use and effectiveness against other toxins/poisons

Since popular sources of chemical poisoning are almost absent in rural areas where the use of herbal antidotes is common, there are very few reports on the use of herbal remedies for many regular poisons and even less evidence of the effectiveness of such uses. Nevertheless, the application and efficacy of reported botanicals against known poisons is the focus of this section. Some naturally occurring plant-derived compounds and extracts have been reported to be useful in counteracting different toxins/poisons, even though certain notorious plant species have also been implicated in food poisoning and other forms of intoxication [95]. Aqueous garlic (*Allium sativum*) extract was found to be a very potent antidote for arsenic poisoning both *in vivo* and *in vitro*, preventing arsenite-induced cytotoxicity, reducing intercellular reactive oxygen species and increasing the activities of antioxidant enzymes. It was also able to counter arsenic mediated incongruity in blood glucose levels [115]. Also, when whole human blood samples were pre-treated with curcumin extracted from *Curcuma longa*, it was found to have a protective effect against radioactive iodine poisoning and had maximal effect at a dose of 50 µg/mL [116].

Ciguatera fish intoxication affects about 100,000 persons annually and is common in the endemic tropical pacific zones [117]. The microscopic algal *Gamberdiscus* species produce the ciguatoxins which accumulate in a variety of fish such as clams, eels, barracuda, parrotfish and grouper, and cause food poisoning to humans consuming such fish. Symptoms of poisoning include vomiting or diarrhoea and dysfunction of the CNS, which present themselves within hours, but can also take up to several months and years. Fortunately, investigators at the Institut de Recherche pour le Développement (IRD) in France came up with a patented antidote after screening of several plant extracts [118]. The octopus bush (*Heliotropium foertherianum*), an ornamental shrub, showed the best conclusive result against cigua toxin fixation. Following several scientific/molecular assays, rosmarinic acid was identified as the active ingredient in *H. foertherianum* extract. Although the compound’s anti-inflammatory, antiviral, antibacterial and antioxidant capacities are well recorded in literature, its use as a detoxifier and antidote for cigua toxin is a relatively current known function. Rosmarinic acid also interestingly serves as a therapy for both the cigua intoxication causes and symptoms. The patented investigation which covers a period of about 20 years serves to cover the development of safe and harmless rosmarinic acid-based derivatives with potentially enhanced detoxification strengths [117,118,119].

Again, plant and animal poisons (leeches, insects, cats, reptiles etc.) are a common problem/challenge in the hilly Mongoloid populated area of Karbi district in Assam State of India where the main occupation is fishing and hunting. Primary healthcare in Karbi is largely covered by folk medicine which inhabitants consider cheap, accessible and safe [95]. The trado-medicinal use of plants in rural and urban areas [120], as well as antidotes sourced from plants [12,113], are well known. In fact, the WHO fosters the integration of trado-medicinal practices into the healthcare system of its member states [120]. Teron and Borthakur [95] reported that plant species within the Zingiberaceae, Solanaceae and Aristolochiaceae families were mostly useful as antidotes against *bab* poisoning in the Karbi area compared to other families such as Acanthaceae, Dioscoreaceae and Oxalidaceae, among others. These were administered in the form of crude extracts, juice, infusions, decoctions, powder, pastes etc. and after derivation from the plants’ leaves, roots, rhizomes, stem, seed, fruit, corm, whole plant, tuber, calyx, bulbil and/or grain. *Hibiscus sabdariffa* leaves and calyx, *Tamarindus indica* and *Citrus* sp. fruits are also useful food poisoning antisera [95]. Recently, a natural nitrophenyl psoralen (NPP) molecule identified from Indian plants and an antidote against *Clostridia botulina* neurotoxin A was shown to possess lower toxicity when tested in an M17 neuroblastoma cellular model [121,122]. The compound actively functioned in preventing the toxin from breaking down and rescuing a synapse-associated protein-25 in the assay cells [122,123]. A narrow range of plant antidotes against snakebite poisons and other selected poisons are further succinctly presented in Table 3.

Furtherance to the foregoing, an attempt was further made to analyse the reported plant parts commonly used as antidotes and this revealed that while the seeds (2.9%) and rhizome (2.9%) found limited usage, the leaves (45.7%) were the most studied and frequently used organ and were used as much as two- and four-fold more than roots and stem bark, respectively (Figure 2). The fruits (8.6%), whole plant (5.7%) and other parts (latex, shoots, calyx and flowers: 7.1%) were intermittently used (Figure 2). Although the frequency of usage of the leafy parts could be closely associated with their ease of accessibility and abundance, this, however, might not be indicative of the leaves eliciting relatively better effectiveness than other parts. The potency of each respective part as an antidote is hugely dependent on the nature and concentration of the resident secondary metabolites. Unfortunately, a good number of the studies lacked detailed characterization of their phytoconstituents.

## 4. Future Prospects And Biotechnological Advancements In Antidote Administration

The record of continued deaths and increased morbidity from drug overdose and other poisons calls for new detox systems development. This has informed studies around the development of antidotal nanomedicine or “nanoantidotes,” with the ability to considerably reverse the toxicity from harmful natural and synthetic xenobiotics in situ and otherwise [46,76]. Such developments have been made possible using biotechnological tools, particularly in the field of nanotechnology. Recently, a nanoantidote in the form of a nanoemulsion/squarticle (containing squalene as the central lipid matrix) was produced and tested for use in the reversal of tricyclic antidepressant (TCA) overdose [7]. Leroux [147] had earlier appraised the important roles that nano-carriers such as nanoemulsions, nanoparticles and macromolecular carriers could play in the delivery of antidotes and in improving the circulation time of bound drugs *in vivo*. In addition, a new *in vivo* screen involving the utilization of the zebrafish was developed in 2011 for detection and diagnosis of OP poisoning. Some of the screen’s identified mode of action include, but are not limited to, reversible AChE inhibition, bioactivation inhibition and cholinergic receptor antagonism [148]. Interestingly, another study added research data to the concept of engineered nanosized click-antidotes which function like a lock-and-key system of enzymes and substrates, but this time between antibodies and antidotes. These would serve in scenarios such as in antibody-induced toxicities so that antibodies are rapidly removed from circulation and tissues by the click-antidote. *In vivo* pharmacokinetic modelling indicated that the click-nanocarriers/antidotes have a rapid click reaction and short half-lives, which lead to an upregulated efficiency in antibody clearance from the blood [149]. Similarly, antidotes for synthetic opioid poisoning, for example, naloxone, have a short pharmacokinetic half-life (30 mins to 1 hour), which is due to its rapid metabolism in the liver, whereas opioid poisoning has a tendency to remain in the circulatory system for over 12 hours, prompting repeated injection of the antidotes at short intervals. A non-toxic biodegradable covalent nanoparticle has been developed using polylactic acid and polyvinyl alcohol. This substance can release sufficient doses of naloxone over a period of 24 hours [150]. Given that naloxone is already proven clinically safe, it was predicted that the product could be available in the market within a short period. These nanoantidote-based investigations set a promising future stage for more advanced development in nanomedicine and indeed, the field of poisoning management/treatment.

Carbon monoxide poisoning is as a result of the formation of a carbon monoxide-hemoglobin (CO-Hb) complex which makes cellular respiration difficult. A neuroglobin compound capable of binding CO about 500 times more than haemoglobin was recently developed and this neuroglobin can clear the CO content of the blood in a very short period of less than a minute. It also increased survival rate, returned heart rate and blood pressure to pre-exposure levels, followed by a rapid renal expulsion of the CO bound neuroglobin in test organisms [151]. It can be administered on the field as opposed to the technical difficulties that comes with the current treatment with hyperbaric oxygen in dive chambers, though the long-term effect of this antidote on neurocognitive dysfunction has not been tested, but there are indications that this could be the future of CO poisoning antidotes. CO is not entirely a gas of no positive use; it has been touted as an important complementary therapy in treating snake envenomation as it was found to significantly inhibit both procoagulant and anticoagulant activities of the hemotoxic venom of serpents of the Elapidae family [87].

Another impressive advancement in this regard is the recent creation of an automated and slightly invasive (subcutaneous) antidote delivery device. The study was informed by more than 40,000 American deaths recorded from opioid poisoning annually and from persons below 50 years of age. The device is inexpensive, simple and in the form of an implantable closed-loop system which dispenses significant doses of naloxone (1.9 mg within 60 s and about 8.8 mg in 600 s) once it detects opioid overdose-induced respiratory failure, thus reducing response time and mortality rate [73]. In another development, RNA-based aptamers/drugs/antidotes which bind specifically to coagulation proteins have been shown to reduce blot clots, which lead to stroke and heart attack [152].

In further discussions from another perspective, the issue of antidote control continues to emerge, but with a positive outlook for the future since protein- and polymer-based molecules which can counteract aptamer activity *in vitro* and *in vivo* are being investigated [153]. In another study, a novel antidote/compound, idarucizumab, was tested as a control for dabigatran, an oral anticoagulant. This was done to proffer a solution should a bleed or anticoagulant overdose scenario arise or to reverse their anticoagulant effect for emergency surgery [154]. Also, further future developments of automated algorithms that give patient-specific antidote treatment, recommendations and individualized dose calculations such the Antidote Application (AA) computational system developed by Long et al. [155] are expected since the AA algorithms only cover about 200 toxins. Research into other modes of antidote administration such as the intraosseous (IO) route utilized in resuscitation scenarios which could enhance antidote efficacy are also ongoing. Such routes would cater for cases of severe poisoning where IV access is unavailable and emergency interventions are required. Studies of IO-administered antidotes are also scarce [156]. There is also a dearth of research data expatiating on antidotes’ toxicity molecular mechanisms/pathways.

More importantly, the role played by traditional medicine cannot also be underscored. The major setbacks in the administration of antidotes in rural areas, which have always been the non-availability of suitable antisera, cost and the stability of the antisera at room temperature, could be easily mitigated by herbal/plant antidotes which are known to transcend these limits, i.e. they are effective for varieties of poisons, relatively cheap and stable at room temperature. Hence, it could only be logical to keep engaging in cutting-edge research to develop novel herbal antidotes by carrying out efficacy and clinical tests to validate their folkloric uses and standardize these invaluable products. The backing of ethnomedicinal antidote knowledge with evidence-based facts of use and efficacy needs to be reemphasized for optimal benefits globally. In determining who gets what antidote, how and when, too often this is based on low quality evidence of risk/benefit assessments [157]. Risk assessments of antidotes need to be reiterated to ensure optimal use of available antidotes for improved benefits (uncertain and time-dependent). This would prevent the unnecessary use and abuse of antidotes by clinicians and/or healthcare professionals who administer them [26,158].

## 5. Conclusions

The estimation of antidote demand remains an arduous task as real-time accurate data on poisoning are not optimally surveyed. Surveillance of poisoning is therefore encouraged, but other factors such as high prices and access to antisera also need to be tackled to ensure adequate management of poisoning cases. A better understanding of the socio-economic impact of poisoning would also facilitate the enforcement of adequate regulation in aspects of clinical toxicology of antidotes and translational research. The utilization of an array of *in vitro* and *in vivo* assays on existing and newly developed antidotes which have surpassed the proof of concept stage, as well as the inclusion of an antidote risk assessment report, would help in providing the required scientific evidence(s) prior to regulatory authorities’ approval. Although adequate measures are available in most developed countries to combat poisoning when reported early, more efficient methods of dissemination and administration of these antidotes are also under appraised. Studies, however, revealed inadequate stocking [5,6,20,21,22] and in some cases, absence of some essential antidotes [4] as major hindrances. This is in sharp contrast to the experience in most rural areas and developing countries where antidotes are hardly available, leaving them with the only option of herbal antidotes. Even though there have been reports on the efficacy of these herbal remedies, the need for safety evaluation and standardization is sacrosanct. Furthermore, an idea of the mechanism behind the antidotal effect of many of these plants have not been adequately elucidated and this is imperative to create a clear path to the standardization of these plant products. The fact that some available ethnobotanical data showed significant efficacy against regular poisons, for example, garlic in arsenic poisoning [115], should be a driver to test them against other regular poisons with a view to advance the field of poisoning. Again, although the development of new antidotal therapies and prophylaxis are encouraged, investigations on the potential short- and long-term complications need to be carried out. Also, with improved follow up, antidote detection probabilities will also be enhanced.

## Figures and Tables

**Figure 1 molecules-25-01516-f001:**
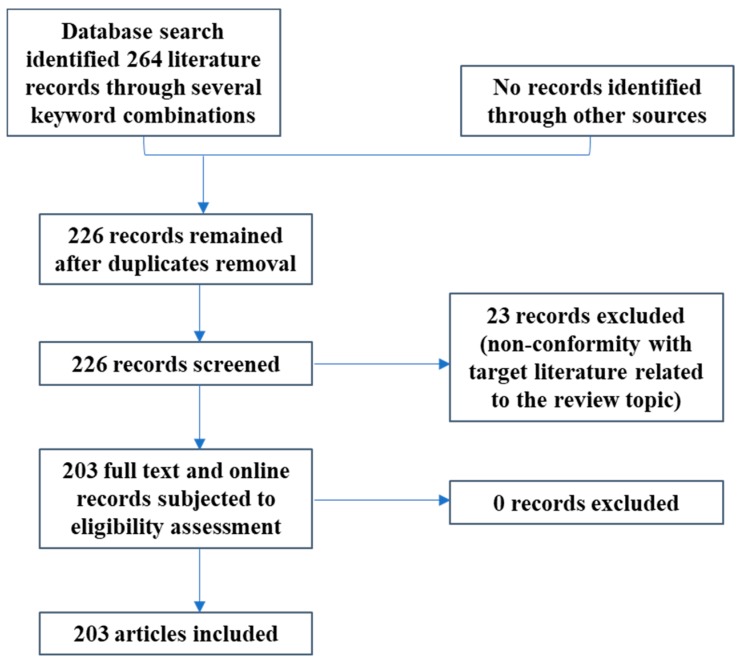
PRISMA flow chart depicting the total of recognized, screened, included and excluded materials for this review.

**Figure 2 molecules-25-01516-f002:**
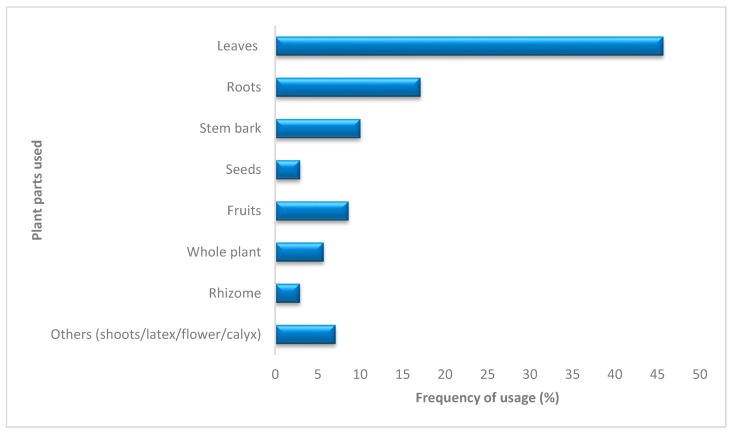
Frequency of plant parts used as antidotes.

**Table 1 molecules-25-01516-t001:** Antidotes and their indications.

Antidote	Poisoning Indication(s)	Reference(s)
Oxygen (Hyperbaric) tetrachloride	Carbon monoxide, cyanide, hydrogen sulfide, carbon	[29,38,39]
Physostigmine	Central anticholinergics	[29]
Potassium hexacyanoferrate, Diethyldithiocarbamate	Thallium	[29]
Succimer (DMSA)	Lead, mercury	[29,30,32]
Dimercaprol	Arsenic, copper, mercury, gold	[29,32,33]
Pyridoxine	Isoniazid, hydrazines, ethylene glycol, gyrometrine	[29]
Flumazenil	Benzodiazepines	[29]
Oximes	Organophosphates	[16,29]
Zwitterionic hydroxyiminoacetamido alkylamines	Organophosphates	[29]
Phytomenadione (Vitamin K)	Coumarin derivatives	[30,41]
Dantrolene	Malignant hyperthermia, Malignant neuroleptic syndrome	[29]
Sodium thiosuphate	Cyanide, bromate, chlorate, iodine	[12,38,39,40]
Activated charcoal	For most poisons, Gastric decontamination	[9,29,47]
Sodium bicarbonate	Metabolic acidosis	[29]
Heparin	Hypercoagulability	[29]
Furosemide	Fluid retention	[29]
Lidocaine	Ventricular arrhythmias	[29]
Promethazine	Allergic reactions	[29]
Copper sulfate	Phosphorus	[29]
Levallorphan, Nalorphine	Opiates	[29]

**Table 2 molecules-25-01516-t002:** Evidence of antidotes’ effectiveness (selected) *in vitro* and *in vivo.*

Experimental Model	Antidote	Indication	Remark	Reference(s)
*In vitro*	DMSA	Mercury poisoning	Moderately effective	[15]
	K203	Tabun gas	Very effective	[17]
	EDTA	Lead poisoning	Moderately effective	[18]
	DMPS	Lead poisoning	Moderately effective	[18]
	DDTC	Lead poisoning	Moderately effective	[18]
	Vitamin B1	Lead poisoning	Moderately effective	[18]
	Vitamin B2	Lead poisoning	Ineffective	[18]
	Vitamin B6	Lead poisoning	Moderately effective	[18]
	Vitamin B12	Lead poisoning	Ineffective	[18]
	Vitamin C	Lead poisoning	Moderately effective	[18]
	Dex40-GTMAC3	UFH	Very effective	[62]
	ATR-4-OX	Paraxon	Moderately effective	[60]
	Activated carbon from *Jatropha curcas*	Zearalenone	Moderately effective	[61]
	Zwitterionic aldoximes	Organo phosphate poisoning	Very effective	[56]
	Obidoxime	Paraxon	Very effective	[59]
*In vivo*	NAC	Mercury poisoning	Very effective	[20]
	SN	Cyanide poisoning	Moderately effective	[21]
	STS	Cyanide poisoning	Moderately effective	[21]
	α-KG	Cyanide poisoning	Very effective	[22]
	K203 + atropine	Tabun gas	Very effective	[17]
	HI-6 + atropine	Soman gas	Very effective	[16]
	Calcium EDTA	Lead/Aluminum poisoning	Very effective	[23]
	Sodium EDTA	Lead/Aluminum poisoning	Moderately effective	[23]
	Cisplatin	Cyanide poisoning	Moderately effective	[69]
	Anionic squarticles	Amitriptyline intoxication	Moderately effective	[72]
	ILE	Cardiotoxicity	Very effective	[74]
	Ethoin	Ricin poisoning	Moderately effective	[79]
	Cobalt-macrocylic compounds	Azide/Cyanide toxicity	Moderately effective	[80]

**Table 3 molecules-25-01516-t003:** Antidotes of plant origin and their induced effect(s).

Poisons	Plant Used	Family	Part Used	Active Metabolite(S)	Mode of Use	Reference(s)
Snake venom	*Bidens Pilosa*	Asteraceae	Leaf	Not available	Leaf juice is applied locally	[95,119,124]
	*Desmodium adscendens*	Fabaceae	Leaf	Triterpenoid aponins Phenylethylamines Indole-3-alkyl amines	Leaf juice is applied locally	[68,95,119,124]
	*Palisota barteri*	Commelinaceae	Stem	Not available	Powdered plant part is applied locally	[95,119,124]
	*Rauvolfia vomitoria*	Apocynaceae	Fruit, leaf and root bark	Not available	Powdered form of all plant parts is applied locally	[95,119,124]
	*Allium sativum*	Liliaceae	Leaf	Allicin Anthocyanine Scordinine A and B	Leaf paste is applied locally	[27,28,95,106,125,126]
	*Eclipta prostate*	Asteraceae	Whole plant	Sitosterol, Stigmasterol D-mannitol Wedelolactone	Not available	[126]
	*Citrus limon*	Rutaceae	Stem bark	4-β-glucopyranoside d-limonene linalcol	Powder of stem bark mixed with water is administered orally	[27,28,95,106,125,126,127,128]
	*Dioscorea alata*	Dioscoreaceae	Whole plant	Not available	Paste of whole plant is applied locally	[27,28,95,106,125,126]
	*Mirabilis jalapa*	Nyctaginaceae	Root	Not available	Extract of root is taken orally	[27,28,95,106,125]
	*Ocimum canum*	Lamiaceae	Leaf	Not available	Leaf paste is applied locally	[27,28,95,106,125]
	*Saccharum bengalense*	Poaceae	Root	Not available	Root is chewed	[27,28,95,106,125]
	*Acalypha indica*	Euphorbiaceae	Root	Terpenoids Phenolic compounds Sterols	Root paste is applied locally	[27,28,95,106,129]
	*Hemidesmus indicus*	Asclepediaceae	Root	2-hydroxy-4-methyl benzoic acid Lupeol acetate	Decoction of root is taken orally	[104,109,111]
	*Achyranthes aspera*	Amaranthaceae	Root	Not available	Root paste is taken with water	[27,28,95,106,125]
	*Calatropis procera*	Asclepidiaceae	Latex	Calotropin Calotropagenin Sterol	Plant latex is applied locally	[27,28,95,106,125,130]
	*Vitex negundo*	Verbenaceae	Leaf	6′-p-hydroxybenzoyl mussaenosidic acid; 2′-p-hydroxybenzoyl mussaenosidic acid Protocatechuic acid; oleanolic acid; flavonoids	Leaf extract is administered orally	[27,28,95,106,125,130,131,132]
	*Piper* spp.	Piperaceae	Seeds	4-nerolidylcatechol	Grinded seed is applied locally	[98]
	*Datura metel*	Solanaceae	Leaf and stem	Meteloidine Hyoscyamine Apoatropine Anisodamine Hyoscine Norharman	Infusion of dry leaf and stem is administered orally	[27,28,95,106,125,133]
	*Curcuma longa*	Zingiberaceae	Rhizome	Turmerin	Powder of rhizome is applied locally	[27,28,95,106,125,134]
	*Azadirachta indica*	Meliaceae	Leaf	AIPLAI	Leaf extract is administered orally	[135]
	*Mimosa pudica*	Mimosaceae	Root	2-Hydroxymethyl- chroman-4-one D-manitolSitosterol	Root extract is administered orally	[27,28,95,106,125,133]
	*Mangifera indica*	Anacardiaceae	Stem bark	Pentagalloyl glucopyranose	Stem bark extract is administered orally	[27,28,95,106,125,136]
	*Aristolochia* spp.	Aristolochiaceae	Leaf	Aristolochic acid	Not available	[28]
	*Hugonia mystax*	Linaceae	Leaf	2-dodecanol Benzene propanoic acid 2-methyl-1-undecanol	Leaf juice is applied locally	[94]
	*Cordia verbenacea*	Borraginaceae	Leaf	Rosmarinic acid	Not available	[28]
	*Mikania glomerata*	Asteraceae	Leaf	Coumarin	Not available	[28]
	*Silybum marianum*	Asteraceae	Root	Silymarin	Root paste is applied locally	[28]
	*Casearia sylvestris*	Salicaceae	Leaf	Ellagic acid	Not available	[137]
	*Symplocos racemosa*	Symplocaceae	Fruits	Benzolsalireposide salireposide	Not available	[28]
	*Cynara scolymus*	Asteraceae	Fruits	Cynarin	Not available	[28]
	*Thea sinensis* (*Camellia sinensis*)	Theaceae	Leaf	Melanin	Decoction of leaf is administered orally	[28]
	*Vernonia condensata*	Asteraceae	Leaf	Caffeic acid	Leaf paste is applied locally	[28]
	*Phyllanthus klotzchianus*	Phyllanthaceae	Whole plant	Quercetin Rutin	Whole plant extract is administered orally	[28]
	*Ceiba pentandra*	Malvaceae	Stem bark	7-hydroxycadalene	Extract of stem bark is administered orally	[94]
	*Sapindus saponaria*	Sapindaceae	Fruits	Flavonoids	Not available	[28]
	*Periandra mediterranea*	Fabaceae	Flower	Triterpenes Sterols Periandrins	Decoction of flower is administered orally	[28]
	*Mandevilla velutina*	Apocynaceae	Leaf	Steroids	Not available	[28]
	*Derris sericea*	Fabaceae	Leaf	Derricidin	Not available	[28]
	*Guiera senegalensis*	Combretaceae	Root	Tannins	Not available	[28]
	*Harpalyce brasiliana*	Fabaceae	Leaf	Edunol	Not available	[28]
	*Dorstenia brasiliensis*	Moraceae	Whole plant	Bergapten	Decoction of whole plant is administered orally	[28]
	*Ehretia buxifolia*	Boraginaceae	Leaf	Ehretianone	Decoction of leaf is administered orally	[28]
	*Derris urucu*	Fabaceae	Leaf	2,5-dihydroxymethyl-3,4-dihydroxypyrrolidine	Not available	[28]
	*Bredemeyera floribunda*	Polygalaceae	Root	Bredemereyosides B and D	Not available	[28]
	*Withania somnifera*	Solanaceae	Root	Indole-3-(4’-oxo) butyric acid	Powder of dry root is applied locally	[94,101]
	*Betula alba*	Betulaceae	Stem bark	Betulin Betulin acid	Decoction of stem bark is administered orally	[28]
	*Tabernaemontana catharinensis*	Apocynaceae	Stem bark	12-methoxy-4-methyl voachalotine	Not available	[138]
	*Pentaclethra macroloba*	Fabaceae	Root	Triterpenoid saponin	Not available	[137]
	*Baccharis trimera*	Asteraceae	Leaf	Neo-clerodane diterpenoid	Leaf paste is applied locally	[139]
	*Pimpinella anisum*	Apiaceae	Leaf	Anisic acid	Not available	[105]
	*Leucas aspera*	Lamiaceae	Leaf	Nerolidol-2	Leaf extract is administered orally	[94]
	*Murraya paniculata*	Rutaceae	Leaf	Trans-nerolidol	Leaf extract is administered orally	[94]
	*Annona squamosal*	Annonaceae	Seed	Eugenol	Seeds are crushed and applied locally	[94]
	*Bixa Orellana*	Bixaceae	Leaf	Germacren-4-ol	Leaf extract is administered orally	[94]
	*Tinospora cordifolia*	Menispermaceae	Leaf	Cordifelone Tinosporidine 3, (a,4-di hydroxy-3-methoxy-benzyl)-4-(4- hydroxy-3-methoxy-benzyl)-tetrahydrofuran	Leaf juice and garlic paste is taken orally	[94,140]
Insecticide/ Pesticide poisoning	*Begonia roxburghii*	Begoniaceae	Whole plant	Not available	Whole plant extract is administered orally	[95]
	*Polygonum affine*	Polygonaceae	Shoot	Not available	Tender shoots are chewed	[95]
	*Polygonum microcephalum*	Polygonaceae	Shoot	Not available	Tender shoots are chewed	[95]
Datura poisoning	*Averrhoa carambola*	Oxalidaceae	Fruits	Phenols Flavonoids	Fruits are eaten	[95,141]
Catfish sting	*Boesenbergia rotunda*	Zingiberaceae	Rhizome	Pinostrobin	Paste of rhizome is applied locally	[95,142]
Bee sting	*Capsicum frutescens*	Solanaceae	Leaf	Capsaicin	Leaf paste is applied locally	[95,143]
	*Lagenaria siceraria*	Cucurbitaceae	Leaf	Not available	Paste of leaf is applied locally	[95]
Food poisoning	*Cayratia pedate*	Vitaceae	Leaf	Triterpenoids Steroids Tannins Phenols	Decoction of leaf is administered orally	[95,144]
	*Xanthium strumarium*	Asteraceae	Leaf	Not available	Leaf juice is administered orally	[95]
Caterpillar poisoning	*Crassocephalum crepidioides*	Asteraceae	Leaf	Ascorbic acid Tannin Saponin Phytate	Paste of leaf is rubbed on the body	[95,145]
Leech bites	*Curcuma longa*	Zingiberaceae	Rhizome	Curcumin	Paste of rhizome is applied locally	[95]
Mushroom poisoning	*Rhus javanica*	Anarcardiaceae	Fruits	Not available	Fruits are chewed	[95]
General chemical poisoning	*Hibiscus sabdariffa*	Malvaceae	Leaf and calyx	Triterpenoids Steroids	Decoction of leaf and calyx is administered orally	[95,146]

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
