# Peer review of "An Appraisal of Antidotes’ Effectiveness: Evidence of the Use of Phyto-Antidotes and Biotechnological Advancements"

_molecules, 2020, doi:10.3390/molecules25071516_

Round 1
Reviewer 1 Report
I found the article well prepared and adequate for publication.
Regarding the following: Literature search was conducted using various electronic data pools such as SciHub, …. I believe that citing a “pirate website in the world to provide mass and public access to tens of millions of research papers” is somewhat strange in this article. I offer the editor in chief the final decision regarding the maintenance of such information. Moreover the following articles could be reviewed for additional discussions. Morgan J. Positivity and pulmonary rehabilitation: antidotes to chronic lung conditions. Lancet Respir Med. 2020. / Frawley KL, Carpenter Totoni S, Bae Y, Pearce LL and Peterson J. A Comparison of Potential Azide Antidotes in a Mouse Model. Chem Res Toxicol. 2020. / Carillo NJ, Golden L and Saraghi M. Flumazenil: a review and implications for benzodiazepine overdose. Gen Dent. 2020;68(1):14-17.
Author Response
Point 1: I found the article well prepared and adequate for publication.
Response 1: Thank you very much for finding the article well prepared and adequate for publication.
Point 2: Regarding the following: Literature search was conducted using various electronic data pools such as SciHub, …. I believe that citing a “pirate website in the world to provide mass and public access to tens of millions of research papers” is somewhat strange in this article. I offer the editor in chief the final decision regarding the maintenance of such information.
Response 2: The statement on SciHub has been revised as recommended
Point 3: Moreover, the following articles could be reviewed for additional discussions. Morgan J. Positivity and pulmonary rehabilitation: antidotes to chronic lung conditions. Lancet Respir Med. 2020. / Frawley KL, Carpenter Totoni S, Bae Y, Pearce LL and Peterson J. A Comparison of Potential Azide Antidotes in a Mouse Model. Chem Res Toxicol. 2020. / Carillo NJ, Golden L and Saraghi M. Flumazenil: a review and implications for benzodiazepine overdose. Gen Dent. 2020;68(1):14-17.
Response 3: We have reviewed the articles and that of ‘Frawlet et al. (2020)’ on azide antidotes has been included in the manuscript (Lines: 348 – 353 and in Table 2). While the work of Morgan (2020) does not entirely fit within the context of our work and hence not included, we have already made mention of and included report on ‘Flumazenil’ in Table 1. We do hope you will find this in order. Thank you very much.
Reviewer 2 Report
Sabiu and co-authors carefully identified 203 articles through various electronic data pools by the keywords and phrases that address the relevance to the scope of this review. In the first part of the review, the authors briefly introduced the importance and significance of antidotes in remedy to poisons. The second part is quite short and states how they conducted the literature search. The third part is the key, which describes ‘Classes of antidotes and mechanism(s) of action’, ‘In vitro studies on the use of antidotes’, ‘In vivo studies on the use of antidotes’, ‘Snake venoms as poisons’ and ‘Natural antidotes of plant origin’. Specifically, the authors reviewed antidote sources, classes, modes of action, the relevance of translational research by identifying valuable plants species used as antidotes. The efficacy of existing antidotes in ‘in vivo’ and ‘in vitro’ investigations is summarised and compared. Following on, they commented on biotechnological advancements in antidote administration, and presented their future prospects. This review is well organized, and makes a significant contribution to the field.
Author Response
Point 1: Sabiu and co-authors carefully identified 203 articles through various electronic data pools by the keywords and phrases that address the relevance to the scope of this review. In the first part of the review, the authors briefly introduced the importance and significance of antidotes in remedy to poisons. The second part is quite short and states how they conducted the literature search. The third part is the key, which describes ‘Classes of antidotes and mechanism(s) of action’, ‘In vitro studies on the use of antidotes’, ‘In vivo studies on the use of antidotes’, ‘Snake venoms as poisons’ and ‘Natural antidotes of plant origin’. Specifically, the authors reviewed antidote sources, classes, modes of action, the relevance of translational research by identifying valuable plants species used as antidotes. The efficacy of existing antidotes in ‘in vivo’ and ‘in vitro’ investigations is summarised and compared. Following on, they commented on biotechnological advancements in antidote administration, and presented their future prospects. This review is well organized, and makes a significant contribution to the field.
Response 1: Thank you very much for recognizing our article as a significant contribution to the field of antidotes.
Reviewer 3 Report
The article titled "An appraisal of antidotes' effectiveness: Evidence of use of phyto-antidotes and biotechnological advancements" provides an interesting review that perfectly fits the aim of the special issue and the journal.
I just have minor comments:
Line 103: Please clarify the use of the terms “efficacy” (under ideal circumstances) and “effectiveness” (under real world clinical settings). The tittle of the article mentions the term “effectiveness” and the aim of the article in the introduction mentions the term “efficacy”. Line 145: The meaning of NAC (N-acetyl cysteine) should be mentioned here because this is the first time that this term is used. Change this concept in line 164. Line 190: I don’t understand the word “nostra”. Line 204: What is the meaning of ST? Line 253: What is the meaning of OP? Sometimes is used the word OP and sometimes is used the words organophosphate poisoning. Line 353: This section is very important, but it should include a section of antidotes used in snake poisoning. The section 3.5 should be used to talk about the antidotes used to other animal venomous poisoning. Line 354: The sentence could be “are classified based on how the effect of their principal toxins are exerted”. Some hemotoxic venoms also induce important neurotoxic effects. Line 385: The meaning of PLA2 should be mentioned here because this is the first time that this term is used. Change this concept in line 437. The number 2 of PLA2 should be written as subscript.
Author Response
Point 1: The article titled "An appraisal of antidotes' effectiveness: Evidence of use of phyto-antidotes and biotechnological advancements" provides an interesting review that perfectly fits the aim of the special issue and the journal.
I just have minor comments:
Line 103: Please clarify the use of the terms “efficacy” (under ideal circumstances) and “effectiveness” (under real world clinical settings). The tittle of the article mentions the term “effectiveness” and the aim of the article in the introduction mentions the term “efficacy”.
Response 1: Efficacy has been changed to effectiveness as used in the title for uniformity. Thank you.
Point 2: Line 145: The meaning of NAC (N-acetyl cysteine) should be mentioned here because this is the first time that this term is used. Change this concept in line 164.
Response 2: Done as suggested.
Point 3: Line 190: I don’t understand the word “nostra”.
Response 3: ‘nostra’ changed to ‘antidotes’.
Point 4: Line 204: What is the meaning of ST? Line 253: What is the meaning of OP? Sometimes is used the word OP and sometimes is used the words organophosphate poisoning.
Response 4: Suggestions amended accordingly.
Point 5: Line 353: This section is very important, but it should include a section of antidotes used in snake poisoning. The section 3.5 should be used to talk about the antidotes used to other animal venomous poisoning.
Response 5: Many thanks for identifying the significance of this section. Section 3.5.1. is specifically on antidotes used for both snake poisoning and those of other animal venomous poisoning. However, the limitation to natural antidotes of plant origin was to be in line with the context of the title of the article which centres mainly on ‘Phyto-antidotes’. Thank you.
Point 6: Line 354: The sentence could be “are classified based on how the effect of their principal toxins are exerted”. Some hemotoxic venoms also induce important neurotoxic effects. Line 385: The meaning of PLA2 should be mentioned here because this is the first time that this term is used. Change this concept in line 437. The number 2 of PLA2 should be written as subscript
Response 6: All corrections effected as recommended.